# HOI4ABOT: Human-Object Interaction Anticipation for Human Intention Reading Collaborative roBOTs

**Esteve Valls Mascaro**[1], **Daniel Sliwowski**[1], **Dongheui Lee**[1,2]

[1] Technische Universität Wien (TU Wien), Autonomous Systems Lab
[2] Institute of Robotics and Mechatronics (DLR), German Aerospace Center
{esteve.valls.mascaro, daniel.sliwowski, dongheui.lee}@tuwien.ac.at
evm7.github.io/HOI4ABOT_page

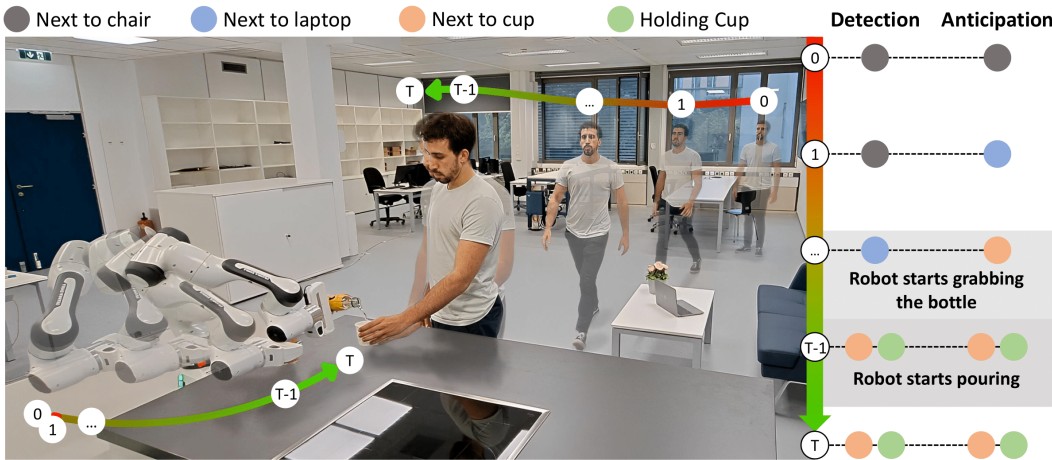

Figure 1: **Overview of our HOI4ABOT framework.** A robot leverages RGB data to detect and anticipate the human-object interactions in its surroundings and assist the human in a timely manner. The robot anticipates the human intention of holding the cup, so it prepares itself for pouring by grabbing the bottle. The robot reacts to the human holding the cup by pouring water.

**Abstract:** Robots are becoming increasingly integrated into our lives, assisting us in various tasks. To ensure effective collaboration between humans and robots, it is essential that they understand our intentions and anticipate our actions. In this paper, we propose a Human-Object Interaction (HOI) anticipation framework for collaborative robots. We propose an efficient and robust transformer-based model to detect and anticipate HOIs from videos. This enhanced anticipation empowers robots to proactively assist humans, resulting in more efficient and intuitive collaborations. Our model outperforms state-of-the-art results in HOI detection and anticipation in VidHOI dataset with an increase of 1.76% and 1.04% in mAP respectively while being 15.4 times faster. We showcase the effectiveness of our approach through experimental results in a real robot, demonstrating that the robot's ability to anticipate HOIs is key for better Human-Robot Interaction.

**Keywords:** Human-Object Interaction, Collaborative Robots, Human Intention

## 1 Introduction

In recent years, the field of robotics has witnessed significant interest in human-robot interaction (HRI), with a focus on enhancing the ability of robots to assist humans in various tasks [1, 2, 3, 4]. To facilitate effective human-robot collaboration (HRC), it is crucial for the robot to possess an

7th Conference on Robot Learning (CoRL 2023), Atlanta, USA.

understanding of both the surrounding environment and the individuals within it, including their intentions. For example, consider the scenario visualized in Fig. 1 where a robot assists a person in the kitchen. By recognizing the person's intention to prepare a drink and understanding their actions such as reaching for the cup, the robot can proactively provide the necessary support in a timely manner, such as picking up a bottle and pouring water. Therefore, by recognizing and anticipating human-object interactions (HOIs), the robot gets a solid understanding of the person's intention and better caters to their needs [1].

While HOI is a long-standing challenge in the computer vision community, most approaches only consider the detection of these interactions from single frames [5, 6, 7, 8, 9, 10]. However, to minimize the latency when a person is assisted by a robot, the detection is not enough, but the anticipation is needed [11, 12, 13]. Therefore, we consider the task of HOI detection and anticipation, and we propose to leverage temporal cues from videos to better understand human intention. HOI recognition in videos has been explored recently [14, 15, 16, 17]. In this paper, we propose a real-time deep learning architecture that combines pre-trained models with spatio-temporal consistency to successfully detect and anticipate HOIs. Our model outperforms the state-of-the-art in VidHOI dataset [14] in terms of accuracy and speed. Moreover, we ensemble our framework with behavior trees [18] to adapt in real-time the robot actions for better interaction with the human. We implement our framework in a real robot and demonstrate the effectiveness of our approach in the pouring task, showcasing the robot's ability to anticipate HOIs and proactively assist the human while reducing latency in the execution.

The contributions of our paper are summarized next:

- A real-time transformer-based model for HOI detection and anticipation.
- A novel patch merging strategy to align image features to pre-extracted bounding boxes.
- To the best of our knowledge, we are the first to assess HOI anticipation in a real robot experiment for a collaborative task.

## 2 Related Works

### 2.1 Human Intention in Robotics

Recognizing and predicting human intention is crucial to ensure seamless human-robot collaboration (HRC) [12, 13, 19, 20]. [12] observed significant differences in the robot's contribution and commitment in an experiment of a human carrying car parts to a shared workspace with an anticipatory robot to assemble them. Recent works in computer vision have highlighted the potential of harnessing human intention to better anticipate future human actions [21, 22, 23]. In particular, [23] leverages the detection of human-object interactions (HOIs) within a scene to understand this high-level intention of the individuals. Despite the benefits of using HOIs, their application in robotics from vision data has not been extensively explored [1]. [4] proposes a conditional random field (CRF) to assess the feasibility of a robot executing a given task based on the anticipated human actions. The CRF predicts the next human actions by considering object affordances and positions in the future. However, [4] is not scalable to new tasks as the CRF relies on hand-crafted features. Instead, we train our model in the largest HOI video dataset available to learn robust features that enhance the robot's ability to anticipate human intention. Recently, [24] proposed a spatial-attention network to extract scene graphs from images in an industrial scenario. However, [24] neglects the time dependency in the task and does not anticipate the human intention to enhance HRC. [25, 26, 27] also adopted scene graphs but focused on task planning.

### 2.2 HOI Detection and Anticipation

HOI focuses on localizing the humans and objects in a scene and classifying their interactions using a ⟨human, interaction, object⟩ triplet (e.g. ⟨person1, hold, cup⟩). HOI task has recently gained attention in the computer vision community due to its promising applications in downstream tasks,

such as scene understanding [28] or action recognition [29]. The primary focus is the detection of HOI from images [5, 6, 7, 8, 9, 10]. Some [7, 8, 9] adopt a one-stage approach, directly operating on the images to predict the HOI triplet. However, these methods require higher training resources and do not benefit from pre-trained object detections. On the contrary, [5, 6, 10] employ a two-stage method to first locate the objects and humans in the image using pre-trained models and then classify each interaction using multi-stream classifiers. In particular, [10] uses a ViT transformer [30] to extract the patched features and proposes Masking with Overlapped Area (MOA) to extract features per object or human through a self-attention layer. Our work shows that weighting the patched features is sufficient to outperform MOA while not requiring any additional parameters.

While processing individual frames may be adequate for HOI detection, we argue that HOI anticipation benefits from leveraging the temporal aspects inherent in these interactions. Several studies in HOI detection address this temporal dimension by focusing on videos [14, 15, 16, 17]. [16] fuses patched features at multiple levels to generate instance representations utilizing a deformable tokenizer. [14] employs a two-stage model that uses 3D convolutions to merge features across the temporal dimension. [15] also adopts a two-stage approach but relies on a spatio-temporal transformer [31] to detect the interactions in videos. Finally, [17] extends the architecture from [15] by concatenating the human and object temporal features and fusing them with the human gaze information using cross-attention. [17] is the first work to propose both HOI detection and anticipation in videos. Similarly to [10], [17] also adopts focal loss [32] to tackle the HOI imbalance in training. We adopt the findings from [17] but observe their model to not be feasible to work in real-time. Moreover, [17] trains a unique model for each anticipation horizon in the future. Instead, we propose a novel real-time multi-head model that can detect and anticipate HOIs in a single step.

### 2.3 Task and Motion Planning

For a robot to effectively assist and collaborate with a human in a particular task, it needs to understand the structure and order of actions involved, enabling the robot to achieve desired goals [33]. Finite State Machines (FSM) have been the standard choice for representing the task structure for a long time [34, 35]. However, scaling FSM poses a challenge due to their lack of modularity and flexibility [18]. Recently, Behavior Trees (BT) [18] have gained popularity as they can facilitate task planning in HRC tasks [36, 37], where the environment is dynamic. Our work adopts BT and defines its behavior based on the anticipated human intention and its uncertainty. Once a suitable chain of actions has been found by the task planner, motion planning is responsible for determining the low-level movements of the robot. Motion planning is a core problem in robotics [38, 39, 40, 41, 42]. [38, 39] proposed to randomly sample points in the state space towards the goal. However, they consider humans as obstacles or constraints, not collaborators. Some approaches [40, 41] formulate motion planning as an optimization problem, but their applications in HRC are limited as determining the cost function related to humans is not trivial. Alternatively, motion generators can be learned from human demonstrations to obtain more natural movement [42, 43]. Dynamic Movement Primitives (DMPs) [42] have been successfully employed in HRC, by dynamically adapting their parameters [44, 45, 46].

## 3 Methodology

In this section, we present our **H**uman-**O**bject **I**nteraction Anticipation for Coll**A**borative ro**BOT**s (**HOI4ABOT**) framework. First, we formulate the HOI detection and anticipation task. Then, we describe the integration of the deep learning architecture into the robot framework.

### 3.1 Human-Object Interaction

Let $\mathbf{V} = [\mathbf{f}_{-T}, \cdots, \mathbf{f}_0]$ be a frame sequence of duration $T + 1$. The goal is to predict the interaction class $i_k^\tau$ in the subsequent time $\tau$ between any human $\mathbb{H}_n$ and object $\mathbb{O}_m$ pair $\mathbb{P}_k = \{\mathbb{H}_n, \mathbb{O}_m\}$ observed during the video $\mathbf{V}$, where $0 \leq n \leq N, 0 \leq m \leq M, 0 \leq k \leq K = M * N$. A visual illustration of our HOI4ABOT architecture is depicted in Fig. 2.

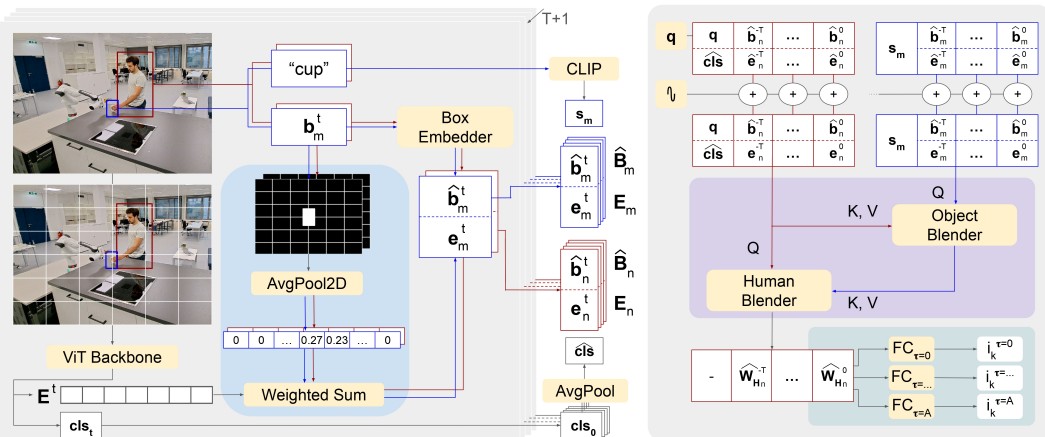

Figure 2: **HOI4ABOT architecture overview.** We consider a video of $T + 1$ frames with the pre-extracted object and human bounding boxes $\mathbf{B}^t$. Our module initially extracts relevant features per frame (left) to later on detect and anticipate HOIs (right) later. First, a ViT backbone [47] extracts patch-based local $\mathbf{E}^t$ and global $\mathbf{cls}_t$ features per each frame $t$. Then, we obtain features per human $\mathbf{e}_n^t$ and object $\mathbf{e}_m^t$ by aligning $\mathbf{E}^t$ to their bounding boxes, as shown in light blue. We also project each $\mathbf{B}^t$ to $\hat{\mathbf{B}}^t$ using a box embedder [48], and the object category to $\mathbf{s}_m$ using CLIP [49]. Our Dual Transformer, shown in purple, leverages the human and object-constructed windows (sequences in red and blue respectively) through two cross-attention transformers, where Key, Query, and Value are used in the attention mechanism. q is a learnable parameter to learn the evolution of the location in time. Finally, we project the enhanced last feature from the Human Blender to detect and anticipate HOIs at several time horizons $i_k^\tau$ in the future through our *Hydra* head (shown in light green).

**Detection and tracking**. HOI4ABOT is a two-stage method. First, we leverage off-the-shelf state-of-the-art object detection and tracking methods to identify the bounding boxes $\mathbf{B}_m \in \mathbb{R}^{(T+1) \times 4}$, label $c_m$, and track identifier $id_m$ for any object $\mathbb{O}_m = \{id_m, c_m, \mathbf{B}_m\}$ in the video $\mathbf{V}$. $\mathbf{B}_m = [\mathbf{b}_m^{-T}, \cdots, \mathbf{b}_m^0]$ represents a list of $XY$ pixel coordinates of the top-left corner and right-bottom corner of the bounding box that locates a given object $\mathbb{O}_m$ at each frame $\mathbf{f}_t$ of $\mathbf{V}$. We obtain the same information for each human $\mathbb{H}_n$. In the second stage, we exploit each individual pair $\mathbb{P}_k = \{\mathbb{H}_n, \mathbb{O}_m\}$ to predict its interaction class $i_k^\tau$ in a given time horizon $\tau$ using various data modalities. This requires understanding the visual features of the pair, how their spatial relationship evolves through time $\mathbf{B}_k = [\mathbf{B}_n, \mathbf{B}_m]$ and also the intrinsic semantics of the object $c_m$.

**Visual features**. We use Dinov2 [47] as a pre-trained Visual Transformer (ViT) [30] backbone to divide each frame $\mathbf{f}_t$ into $L \times L$ patches and project each patch $\mathbf{p}_l^t$ to a visual token $\mathbf{e}_l^t$ that encodes the image information of that patch $l$. In total, the image encoder obtains $\mathbf{E}^t \in \mathbb{R}^{L^2 \times d}$ that captures the local visual features, plus the global context vector $\mathbf{cls}_t \in \mathbb{R}^d$ of a frame $\mathbf{f}_t$.

We develop a simple but efficient technique, called Patch Merger, to extract individual features per human and object from a frame through a single step. Let $\mathbb{O}_m^t$ be an object $m$ with its box $\mathbf{b}_m^t$ at frame $\mathbf{f}_t$. First, we create a binary mask for $\mathbf{f}_t$, where 1 denotes a pixel laying within $\mathbf{b}_m^t$. We convert the binary mask in a sequence of patches following [30]. Then, we obtain a weighting vector $\boldsymbol{\omega}_m^t$ by computing the percentage that $\mathbf{b}_m^t$ overlaps each patch using 2D Average Pooling and normalization. Finally, we compute the weighted sum of local visual features $\mathbf{e}_m^t = \sum \boldsymbol{\omega}_m^t \mathbf{E}^t$, obtaining the individual representation of $\mathbb{O}_m^t$. Compared to [10], which normalizes along the patch dimension and uses a quantized sequence as the attention mask for a self-attention layer, our algorithm is parameter-free, more efficient, and shows better performance in our experiments.

We propose to capture the context within a frame using $\mathbf{cls}_t \in \mathbb{R}^d$, contrary to the spatial transformer proposed in [17]. We claim that this context (e.g. a kitchen, an office) should be invariant in short time periods and be the dominant component among all $\mathbf{cls}_t$ tokens. Consequently, we use Average Pooling to reduce the N $\mathbf{cls}_t$ features to a single representation $\widehat{\mathbf{cls}} = AvgPool([\mathbf{cls}_{-T}, \cdots, \mathbf{cls}_0])$, which is the context of the scene.

**Spatial features**. For each bounding box $\mathbf{b}_m^t$, we extract the $XY$ normalized pixel coordinates for the top-left corner and right-bottom corner. Then, we adopt a positional encoding using random spatial frequencies [48] to embed the location of each point and merge these two corner representations into one box representation $\hat{\mathbf{b}}_m^t \in \mathbb{R}^d$ using a fully connected layer. This process is also applied to humans, thus obtaining $\hat{\mathbf{b}}_n^t \in \mathbb{R}^d$ to encode each human $\mathbb{H}_n^t$ position in the scene.

**Object semantics**. Leveraging the object semantics is essential to understanding the possible interactions in a given pair. While 'holding a cup' or 'holding a bottle' are both feasible, 'holding a car' becomes more unrealistic. Thus, we extract object semantic information $\mathbf{s}_m \in \mathbb{R}^d$ per object $\mathbb{O}_m$ to facilitate the model predicts the intention class $i_k^\tau$. For that, we use the CLIP text encoder [49].

**Pair Interaction.** We construct a temporal architecture that leverages the evolution of the interactions between a human $\mathbb{H}_n$ and an object $\mathbb{O}_m$ in time. We process each pair independently, and therefore we focus on a single pair in the formulation. We stack both the visual tokens $\mathbf{E}_n = [\mathbf{e}_n^{-T}, \cdots, \mathbf{e}_n^0]$ and the spatial features $\hat{\mathbf{B}}_n = [\hat{\mathbf{b}}_n^{-T}, \cdots, \hat{\mathbf{b}}_n^0]$ in time and construct a human temporal window $\mathbf{W}_{\mathbf{H}n} = [\hat{\mathbf{B}}_n, \mathbf{E}_n]$. Similarly, we also construct an object's temporal window $\mathbf{W}_{\mathbf{O}m} = [\hat{\mathbf{B}}_m, \mathbf{E}_m]$. We add a sinusoidal positional encoding to $\mathbf{W}_{\mathbf{H}n}$ and $\mathbf{W}_{\mathbf{O}m}$, Later, we prepend the global visual feature and a learnable spatial parameter $[\mathbf{q}, \widehat{\mathbf{cls}}]$ to $\mathbf{W}_{\mathbf{H}n}$. $\mathbf{q}$ learns the evolution of the location of the human in time through the attention mechanism. We also extend $\mathbf{W}_{\mathbf{O}m}$ by prepending the semantic token $\mathbf{s}_m$ that encodes the object type. Therefore, we obtain a temporal feature $\mathbf{W}_{\mathbf{H}n} \in \mathbb{R}^{(T+2)\times d}$ and $\mathbf{W}_{\mathbf{O}m} \in \mathbb{R}^{(T+2)\times d}$ per pair.

To extract the HOI relationships between $\mathbb{H}_n$ and $\mathbb{O}_m$, we train a dual transformer with cross-attention layers. First, an Object Blender transformer enhances the object window $\mathbf{W}_{\mathbf{O}m}$ based on the human knowledge $\mathbf{W}_{\mathbf{H}n}$. Then, the blended object features $\widehat{\mathbf{W}_{\mathbf{O}m}}$ are used to extend the human representation $\mathbf{W}_{\mathbf{H}n}$ in the Human Blender transformer to $\widehat{\mathbf{W}_{\mathbf{H}n}}$. Finally, we extract the last token from $\widehat{\mathbf{W}_{\mathbf{H}n}}$, which encodes the most current status of the scene, and classify the interaction pair $i_k^\tau$ using a fully connected layer. As a given human-object pair can have multiple interactions simultaneously, we use a sigmoid function and define a threshold to classify the current interactions.

**Multi-head classification for multiple future horizons.** The goal is to predict the interaction class $i_k^\tau$ in the subsequent time $\tau$ between any human $\mathbb{H}_n$ and object $\mathbb{O}_m$ pair $\mathbb{P}_k = \{\mathbb{H}_n, \mathbb{O}_m\}$. We considered the problem of HOI detection ($\tau = 0$) and also the anticipation in multiple future horizons ($\tau > 0$). Contrary to [17] that proposes one trained model for each $\tau$, we developed a single model that can predict multiple time horizon interactions. For that, we froze the HOI4ABOT trained in the detection task, and train an additional linear layer that projects the last token from $\widehat{\mathbf{W}_{\mathbf{H}n}}$ to the interaction for the particular $\tau$. We call this shared backbone the *Hydra* variant, which allows us to simultaneously predict interactions across multiple $\tau$, making our model faster and more efficient. We consider our *Hydra* variant with $A$ number of heads.

## 3.2 Motion generation and task planning

**Motion Generation.** The proposed framework segments the complex movements into simpler movement primitives, which are learned with DMPs. To collect demonstrations of each movement primitive, we employ kinesthetic teaching, where an operator guides the robot's end effector by physically manipulating it [50]. Generating the motion requires estimating the goal position, which we obtain through the use of a calibrated vision system that relies on a pre-trained object detector (i.e. YOLOv8 [51]) and a depth camera. The position of the goals with respect to the robot base is computed using the intrinsic and extrinsic camera matrices.

**Task planning.** Properly scheduling the acquired movement primitives is crucial to reach a desired goal. We implement Behavior Trees (BT) [18] as a ROS node that subscribes to the predicted HOIs and their confidence. The reactiveness of BTs allows adapting the robot's behavior by considering the anticipated human intention and changing to the appropriate sub-tree if needed. This is motivated by how humans interact with each other. For example, if a bartender observes a client approaching the bar, they can prepare for the interaction by grabbing a glass, thus reducing the serving time.

<table>
</table>

| Table 1: Detection mAP. | | | |
|---|---|---|---|
| Method | Full | Non-Rare | Rare |
| **Oracle Mode** | | | |
| ST-HOI [14] | 17.6 | 27.2 | 17.3 |
| QPIC [54] | 21.4 | 32.9 | 20.56 |
| TUTOR [16] | 26.92 | 37.12 | 23.49 |
| STTran [15] | 28.32 | 42.08 | 17.74 |
| ST-Gaze [17] | 38.61 | 52.44 | 27.99 |
| Ours (*Dual*) | 40.37 | **54.52** | 29.5 |
| Ours (*Stacked*) | **40.55** | 53.94 | **30.26** |
| **Detection Mode** | | | |
| STTran [15] | 7.61 | 13.18 | 3.33 |
| ST-Gaze [17] | 10.4 | 16.83 | 5.46 |
| Ours (*Dual*) | **11.12** | **18.48** | **5.61** |
| Ours (*Stacked*) | 10.79 | 17.79 | 5.42 |

Table 2: Anticipation mAP in Oracle mode.

| Method | $\tau_a$ | mAP | Preson-wise top-5 | | | |
|---|---|---|---|---|---|---|
| | | | Rec | Prec | Acc | F1 |
| STTran [15] | 1 | 29.09 | **74.76** | 41.36 | 36.61 | 50.48 |
| | 3 | 27.59 | **74.79** | 40.86 | 36.42 | 50.16 |
| | 5 | 27.32 | **75.65** | 41.18 | 36.92 | 50.66 |
| ST-Gaze [17] | 1 | 37.59 | 72.17 | 59.98 | 51.65 | 62.78 |
| | 3 | 33.14 | 71.88 | 60.44 | 52.08 | 62.87 |
| | 5 | 32.75 | 71.25 | 59.09 | 51.14 | 61.92 |
| Ours (*Dual, Scratch*) | 1 | **38.46** | 73.32 | 63.78 | 55.37 | 65.59 |
| | 3 | 34.58 | 73.61 | 61.7 | 54 | 64.48 |
| | 5 | 33.79 | 72.33 | 63.96 | 55.28 | 65.21 |
| Ours (*Dual, Hydra*) | 1 | 37.77 | 74.07 | **64.9** | **56.38** | **66.53** |
| | 3 | 34.75 | 74.37 | 64.52 | 56.22 | 66.4 |
| | 5 | 34.07 | 73.67 | 65.1 | 56.31 | 66.4 |

**Robot control.** The generated poses from the motion generator are passed to the controller. In our system, we employ a Cartesian impedance controller [52, 53] to achieve the compliant behavior of the manipulator. This controller enhances the safety of human-robot collaboration by allowing the robot to respond in a compliant manner to external forces and disturbances.

# 4 Experiments

## 4.1 Dataset and Metrics

We train and evaluate our model on the VidHOI dataset [14], the largest dataset available for human-object interactions in videos. This dataset encompasses 7.3 million frames with 755,000 annotated interactions of one frame per second. To assess the performance of our approach, we adopted the same evaluation metrics as those presented in [17]. We computed the mean average precision (mAP) using the method presented in [54]. The mAP@50 incorporates the precision-recall curves for all interaction classes. To determine a correct HOI triplet, three conditions need to be met: (i) the detected bounding boxes for human and object must overlap with their corresponding ground truths with an Intersection over Union (IoU) of 50 %, (ii) the predicted object category is correct, (iii) the predicted interaction is correct. Following standard evaluation in VidHOI, we report mAP across three different HOI sets: (i) Full: all interaction categories, (ii) Non-Rare: frequent interactions in the validation set (more than 25 appearances), (iii) Rare: non-frequent interactions (less than 25). Additionally, we evaluated our approach in *Oracle mode*, where we use the human and object detections from ground truth, and in *Detection mode*, where those are predicted using YOLOv5 [55] as in [17]. Finally, we computed the Person-wise top-k metrics [17] where the anticipation was considered correct if one of the top-k predicted interactions matched the ground truth.

## 4.2 Quantitative evaluation

HOI4ABOT outperforms state-of-the-art models [14, 54, 16, 15, 17] in terms of accuracy and speed across all different tasks and scenarios, as shown in Table 1 and Table 2. Moreover, Table 2 shows how our *Hydra* variant outperforms all models in the anticipation task, even training from scratch a separate model for each anticipation horizon. We consider that the detections provide a great deal of information regarding what a human is doing now, and what they might be interested in doing next. By using the *Hydra* variant we ground the anticipation to what is happening at the present time.

## 4.3 Ablation study

This section analyses our proposed approaches and their impact on the performance of the HOI task. All results are depicted in Table 3. For simplification, we only consider the HOI detection task.

Firstly we explore different variations in the extraction and arrangement of features to compose the human and object windows. We compare our Patch Merger strategy to the MOA strategy from [10]. Using MOA requires an additional self-attention block, which increases the model's parameters while underperforming. Moreover, we explore different feature aggregation strategies to classify an interaction. Instead of using the last observed token in $\widehat{\mathbf{W}}_{\mathbf{H}n}$ for classification, we prepend an additional learnable token to $\mathbf{W}_{\mathbf{H}n}$ which aggregates the interaction relationships, inspired by the ViT class token [30].

Table 3: Albation study in HOI detection.

| Variant | mAP |
|---|---|
| Feature blender = *MOA* | 40 |
| Interaction token = *Learnable* | 40.29 |
| Main branch = *Object* | 39.85 |
| Transformer type = *Single* | 40.26 |
| Transformer type = *Stacked* | **40.55** |
| *Dual* | 40.37 |

However, Table 3 shows that classifying from the last observed features is better while not requiring additional parameters. Last, we consider varying the order of the cross-attention branches, first the Human Blender and second the Object Blender. We claim that the decrease in performance is due to the different behavior between humans and objects: objects are static and therefore less informative than humans, which are dynamic and lead the interaction.

Secondly, we assess our dual transformer by comparing it with other variants. We consider the *Single* variant when only using the Human Blender transformer, which is not able to effectively capture the HOIs. We also consider stacking both $\mathbf{W}_{\mathbf{H}n} \in \mathbb{R}^{(T+2)\times d}$ and $\mathbf{W}_{\mathbf{O}m} \in \mathbb{R}^{(T+2)\times d}$ to a single feature window pair, $\mathbf{WP}_k \in \mathbb{R}^{(T+2)\times 2d}$. We observe slight improvements in this variant in terms of mAP when detecting in the *Oracle mode*, but it underperforms in the *Detection mode* and for the anticipation tasks, as shown in Appendix E.

Finally, we compare the inference time of our model to [17] to assess the efficiency in real-world applications in robots. Our *Dual* variant is 15.4 times faster than [17] for the detection task. [17] requires extracting gaze maps, which drastically slows down the inference speed of their model. When using our *Hydra* model, we obtain interactions for the time horizons 0, 1, 3, and 5 using one forward pass, with nearly the same inference speed and parameters as using one head. More information can be found in Appendix D.

## 4.4 Real World Experiments

HOI detection and anticipation are essential for robots to comprehend the surrounding humans and better predict their needs, so the robot can assist in a timely manner. We conduct real experiments with a Franka Emika Panda robot to showcase the benefit of our approach in collaborative robots beyond the offline VidHOI dataset. The VidHOI dataset contains user-collected videos of humans, mostly performing outdoor activities that can not be easily related to robotic collaboration tasks. We consider the 'pouring task' in a kitchen scenario where the robot assumes the role of a bartender with the goal of pouring a beverage for the human. The scenario is shown in Fig. 1. To assess the performance of our model in unseen scenarios, we collected 20 videos of 5 people in our kitchen lab. The human is instructed to grab the cup and informed that the robot will assist them in the task. We manually annotate the time the person grabs the cup to use as ground truth. Our *Hydra* variant detects and anticipates the HOI between a person and a cup in real-time. When the robot anticipates that the human will be near the cup, it proceeds to grab the bottle. However, if the human moves away the robot releases the bottle and returns to the initial pose. The robot proceeds to pour the liquid into the cup after detecting that the human is holding it.

We assess our real-world experiments by considering well-established metrics in HRC [13]. [13] proposes to evaluate human-robot fluency in the joint task by considering four objective metrics. *Human Idle Time* (H-IDLE) and *Robot Idle Time* (R-IDLE) are proposed to evaluate the percentage of the total task time that the respective agent is not active, which reflects the team coordination and the inefficiency of the agent in the task. *Concurrent Activity* (C-ACT) measures the percentage of total task time in which both agents are active concurrently (the action overlap between different members). A higher C-ACT indicates a better-synchronized team. *Functional Delay* (F-DEL) measures the delay experienced by the agents immediately after completing an activity: the percentage

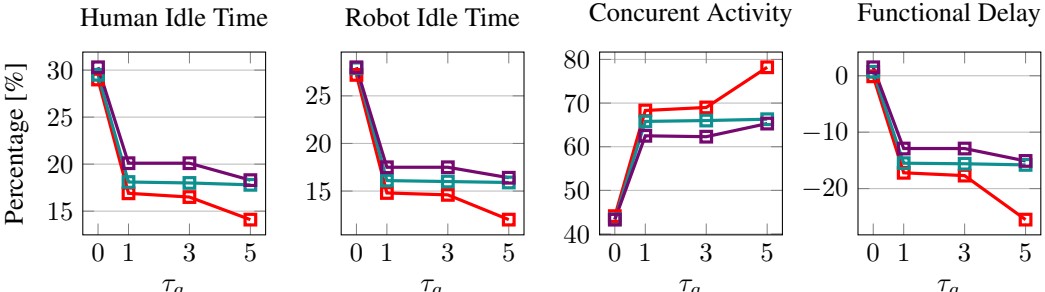

Figure 3: Mean objective fluency metrics for pouring experiments for different confidence thresholds $\{0.3, 0.5, 0.7\}$ in the HOIs prediction.

of total task time between the completion of one agent's action and the beginning of the other agent's action. A negative F-DEL indicates that actions are overlapping and implies an efficient use of team members' time. Figure 3 summarizes the average objective fluency metrics across our pouring experiments. The results indicate that HOI anticipation allows for better human-robot coordination and efficiency of each other's time, thus making the task more fluent. We observe a substantial improvement in Figure 3 when using anticipation ($\tau_a > 0$) compared to detection ($\tau_a = 0$). Additional quantitative and qualitative results are provided in Appendix B.

## 5    Limitations

Despite outperforming state-of-the-art models in HOI from videos, we observe from qualitative experiments the challenge of the implementation in the real world. First, there is a domain gap between the VidHOI dataset, mainly representing humans in daily scenes and our robotic scenario. For instance, anticipating that 'a human is holding a cup' is challenging, despite being correctly detected. We explore the VidHOI dataset and observe that most people already appear with the cup in their hand. To overcome this issue, we sample more frequently clips where the interaction changes in the anticipation horizon. Still, this is insufficient to ensure correct anticipation in 'holding a cup' with higher confidence. Other datasets are not better suited for our problem as they mainly are image-based [5, 56] or do not track the humans and objects in videos [57]. Future research directions consider training with a dataset more coupled to our robotics scenario to improve the model predictions. This would allow us to extend our experiments to more complex daily scenarios. Second, in our real experiments, we assume that the objects present in the scene are sufficiently visible so that object detection can recognize them. Finally, the employed DMPs could be expanded or replaced by visual servoing to consider goal-following behaviors.

## 6    Conclusions

In this paper, we proposed a **H**uman-**O**bject **I**nteraction Anticipation for Coll**A**borative ro**BOT**s framework (**HOI4ABOT**). We consider the task of detecting and anticipating human-object interactions (HOI) in videos through a transformer architecture. We train and evaluate HOI4ABOT in the VidHOI dataset and outperform current state-of-the-art across all tasks and metrics while being $15.4\times$ faster. Moreover, our model runs in real-time thanks to our efficient design. Additionally, we extend our HOI4ABOT model with a multi-head architecture, which can detect and anticipate HOIs across different future horizons in a single step. We demonstrate the effectiveness of our approach by implementing our model in a Franka Emika Panda robot. We show that anticipating HOIs in real-time is essential for a robot to better assist a human in a timely manner and we support our findings with real experiments. In conclusion, our approach demonstrates its effectiveness and defines a new road to explore, where intention reading plays a crucial role for robots in collaboration scenarios.

**Acknowledgments**

This work is funded by Marie Sklodowska-Curie Action Horizon 2020 (Grant agreement No. 955778) for the project 'Personalized Robotics as Service Oriented Applications' (PERSEO).

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

# Appendix

## A  Implementation Details

In this section, we offer a comprehensive summary of the implementation details to aid in the reproduction of the experiments and the replication of the results. All experiments were conducted using a single NVIDIA RTX A4000 graphics card with 16GB of memory and an Intel i7-12000K CPU.

**Hyperparameters.** All trained models are conducted using the same strategy as [17]. We use the official code from https://github.com/nizhf/hoi-prediction-gaze-transformer and implement our HOI4ABOT model into their framework. All training settings are summarized in Table 4. We adopt Cross Binary Focal Loss [32] with $\gamma = 0.5$ and $\beta = 0.9999$, which improves training in extremely imbalanced datasets, such as VidHOI [14]. We train our models using the AdamW optimizer [58]. We define a scheduler for the learning rate, with an initial value of $1 \times 10^{-8}$ that increases to a peak value of $1 \times 10^{-4}$ in 3 warm-up epochs. The learning rate then decreases with an exponential decay with a factor $0.1$. We run the training for 40 epochs.

**Model configuration.** All trained models use a similar configuration, but some variants such as *Stacked* or *Single* are adapted to ensure having a similar number of trainable parameters in the architecture (57.04M). All models reported in our paper use the DINOv2 [47] as the image feature extractor, using the smallest variant available ViT-B/14 that only contains 22.06M parameters; and CLIP [49] for the semantic extractor, with the largest available variant ViT-L/14 that contains 85.05M parameters. However, due to the fact that the number of objects in the dataset is limited, we pre-extracted the features for all possible objects. For our baseline HOI4ABOT model, we consider two transformer models with cross-attention layers, each of them with depth 4 and MLP expansions of ratio 4.0. Each transformer uses the multi-head attention variant with 8 heads to better extract the relationships within a sequence of features. Moreover, we consider sinusoidal positional embedding to facilitate learning the temporal information of a sequence. Finally, we consider the embedding size of each extracted feature, bounding box, or image feature, as 384. The embedding size for the prepended class token is also 384, as this is the embedding dimensions of the features extracted using DINOv2. For the semantics, CLIP obtains a feature of dimensionality 764.

| Table 4: Training settings. | |
|---|---|
| Optimizer | AdamW |
| Weight Decay | 1.0e-2 |
| Scheduler | ExponentialDecay |
| Warmup Epochs | 3 |
| Initial LR | 1e-8 |
| Peak LR | 1e-4 |
| Exponential Decay | 0.1 |
| Epochs | 40 |
| Random Seed | 1551 |
| Augmentation | Horizontal Flip |
| Flip Ratio | 0.5 |
| Batch Size | 16 |
| Dropout | 0.1 |

| Table 5: Model settings. | |
|---|---|
| Transformer Depth | 4 |
| Number of Heads | 8 |
| Feature Extractor | DINOv2: ViT-B/14 [47] |
| Semantic Extractor | CLIP: ViT-L/14 [49] |
| Embedding Dimension | 384 |
| Positional Embedding | Sinusoidal |
| Exponential Decay | 0.1 |
| Mainbranch | humans |
| MLP ratio | 4.0 |

## B  Experimental Scenario

**Task description.** Our HOI4ABOT framework enhances human intention reading through HOI anticipation. We conduct a real-world experiment with a Franka Emika Panda robot to support our proposed approach. Fig. 4 provides a step-by-step overview of the considered bartender scenario. First, the robot detects a human in the scene and anticipates the human intention to approach a kitchen island. When the robot anticipates with confidence that the human will be close to the cup,

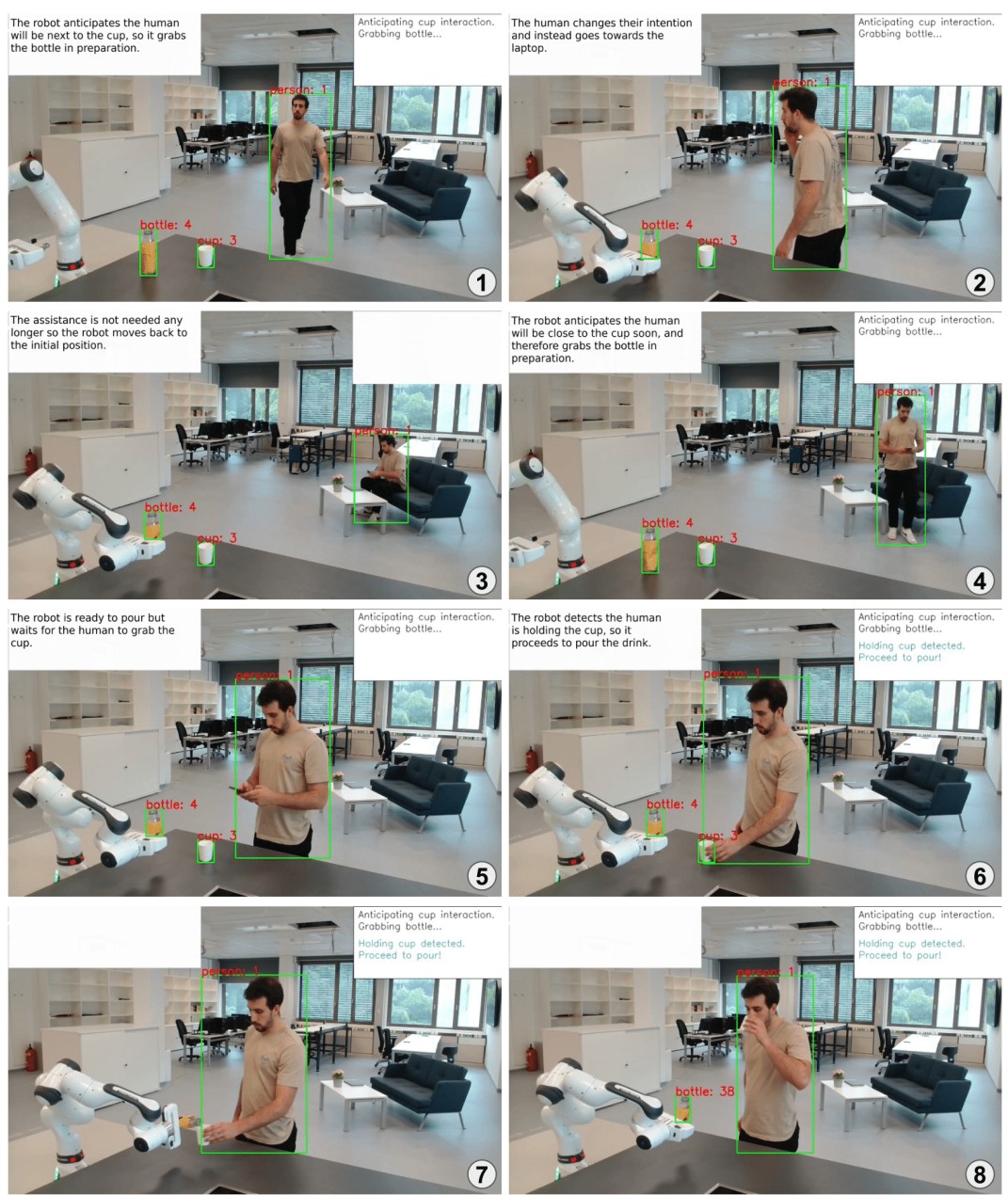

Figure 4: Real-world experiments scenario.

it executes a movement to grab the bottle, thus preparing for pouring. If the intention of the human changes, the robot adapts its behavior and moves back to the initial position after placing the bottle on the table. on the other hand, if the human proceeds to grab the cup, the robot pours the drink and goes back to its initial position. This preparatory behavior reduces the serving time while improving the overall experience for the human.

**Additional Qualitative Results** On the project website (https://evm7.github.io/HOI4ABOT_page/) we present additional qualitative results that showcase the ability of our model to operate in more ambiguous scenarios (with multiple objects and people instances, and cluttered scenes) and execute different motions depending on the predicted interactions. Our model predicts the interaction associated with specific human and object instances, which are associated with the identifiers obtained from the tracker. Therefore, we are able to execute different movements depending on the

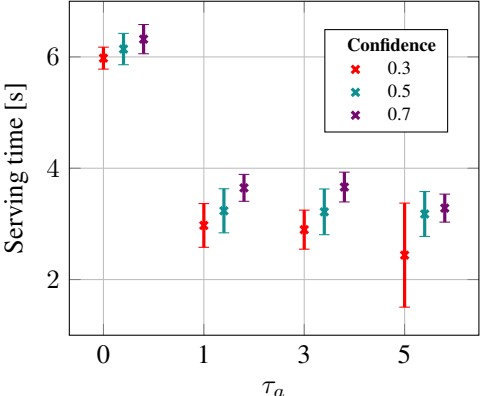

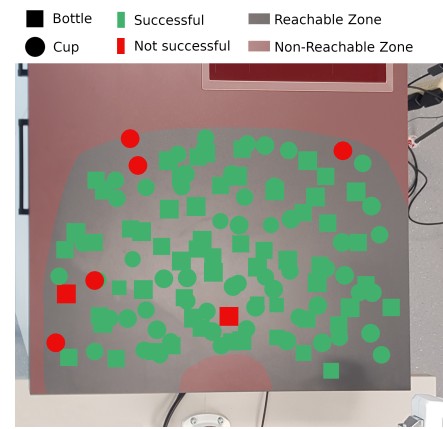

Figure 5: Human waiting time to be served the drink for different confidence thresholds ($\{0.3, 0.5, 0.7\}$) and anticipation heads $\tau_a = \{0, 1, 3, 5\}$.

Figure 6: **Quantitative evaluation of the pouring task.** We overlay on the image of the workspace of the robot the position of the bottle and the cup. Green signifies successful task execution and red failed cases.

interaction (like pouring, pushing, or turning off the lights) and the object category (grabbing from the side in case of a cup or a bottle or grabbing from the top in case of a bowl) or instance (cup-1 and cup-2). For instance, we consider the case of multiple cups in the scene, where the robot conditions its pouring behavior based on the cup the human holds. Additionally, we also show the ability to operate in an ambiguous situation with multiple objects and people instances.

**Evaluation of the use case scenario of HOI4ABOT**. To validate our hypothesis, we extend our evaluation of the framework in real-world experiments with additional quantitative metrics. First, we assess the human waiting time until the robot proceeds to serve. Fig. 5 shows the quantitative benefit of our approach by considering the absolute time a human waits to be served (serving is considered until the robot starts pouring). The results indicate that our robot behaves proactively when anticipating HOIs and therefore reduces the time to wait until a drink is poured, compared to the reactive behavior observed if the robot is only detecting HOIs. Fig. 5 shows a slight reduction in the waiting time when reducing the confidence threshold in the prediction: to be more confident in the human intention the robot waits more. In addition, we observe only a slight decrease in the waiting time for different anticipation horizons ($\tau_a = \{1, 3, 5\}$). This subtle variation might be caused because of the dataset limitation pointed out in the main manuscript. Secondly, we measure the effectiveness of our robot pouring a drink in our real-world trials by considering the success rate of the pouring task in 20 new real-world experiments. Four lab members were instructed to approach the robot and grab the cup. Each person did 5 repetitions. Our framework correctly executes the pouring task in 17 out of 20 executions, resulting in a success rate of $85\%$.

## C   Motion Generation and Task Planning

**Motion Generation.** Our framework decomposes the complex movements into simpler movement primitives, which are learned with DMPs. For instance, the pouring task consists of multiple steps, like grabbing the bottle, moving to the cup, tilting the bottle, and placing the bottle back. Learning the entire movement as a single primitive is possible, but this might oversimplify the motion, particularly for sharp movements, compromising accuracy. In our experiments, each motion segment was learned from a single demonstration. We verified the success rate in the pouring scenario with 60 different arrangements of the objects 'Bottle' and 'Cup'. Figure 6 in the attached document shows that our robot successfully pours 53 out of 60 executions, resulting in a success rate of $88.33\%$. We can observe that failure cases occur mainly when the objects are arranged close to the non-reachable

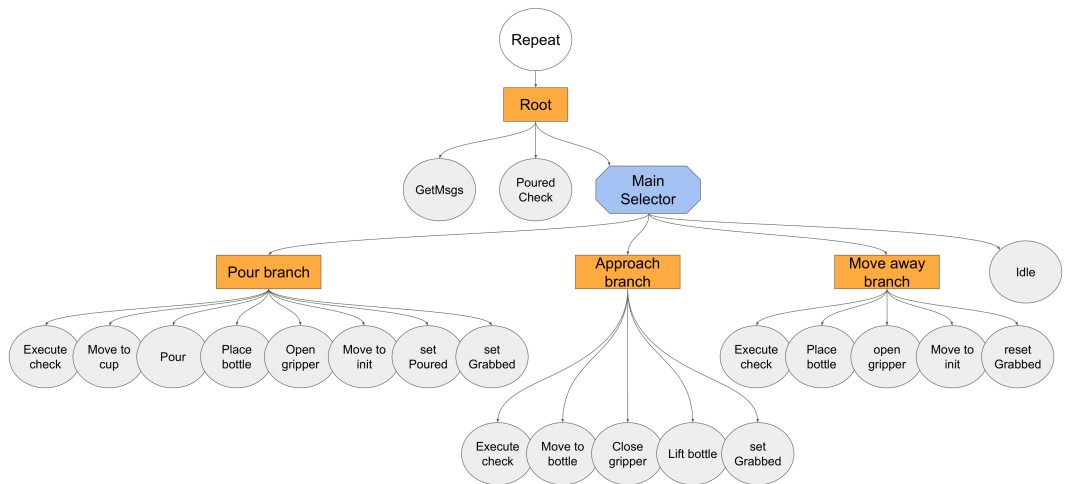

Figure 7: Schematic of the Behaviour Tree for our HOI4ABOT framework.

areas for the robot. Working close to the non-reachable zone is more problematic as the robot is operating near its kinematic constraints. However, this issue can be solved by rearranging the robot base position adequately to the user's need.

**Task Planning: Behavior Tree.** In this section, we describe the structure of the Behavior Tree [18] used in our real-world experiments, which is shown in Fig. 7. The primary focus of this work is to enhance human-robot collaboration through human intention reading using HOI anticipation. We conduct a simple real-world experiment with a Franka Emika Panda robot to showcase the benefits of our approach. This paper does not intend to provide a general development of BT for HOI tasks. However, the same methodology employed can be extended to more complex scenarios thanks to the modularity of BT.

The entire tree is built from three sub-trees: the *Pour branch*, the *Approach branch*, and the *Move Away branch*. First, the *Pour branch* is responsible for pouring the liquid into the cup. It is executed once the bottle is grabbed, and the 'hold' interaction between the human and the cup is detected. To achieve this conditional execution we add the *Execute check* behavior at the beginning of the branch. Then, we reset the *Grabbed flag* and set the *Poured flag* to prevent any potential duplication of pouring into the cup. Secondly, the goal of the *Approach branch* is to grab the bottle. This sub-tree is executed when the bottle is not currently grabbed and the robot anticipates the 'next to' interaction with a confidence greater than a pre-defined threshold. Once the bottle is grabbed, the *Grabbed flag* is set. Thirdly, the *Move Away branch* is responsible for releasing the bottle and moving it back to its initial position. This branch is executed when the bottle is grasped by the robot and the robot anticipates the interaction 'next to' with a confidence lower than a predefined threshold. After executing the movements the *Grabbed flag* is reset.

The appropriate sub-branch is selected by using the *Main Selector* composite node. This node attempts to execute each sub-tree starting from left to right. The selector node executes the next branch in the sequence when the check in the preceding branch is not satisfied. Finally, the last behavior in the sequence is an *Idle* behavior where the robot waits for a short period of time.

The root of the tree is a sequential node, which first collects all messages from the appropriate ROS topics, next checks if the beverage has been already poured, and finally executes the *Main Selector*. To achieve continuous operation, the *Root* node is decorated by a *Repeat* modifier, which executes the root node indefinitely.

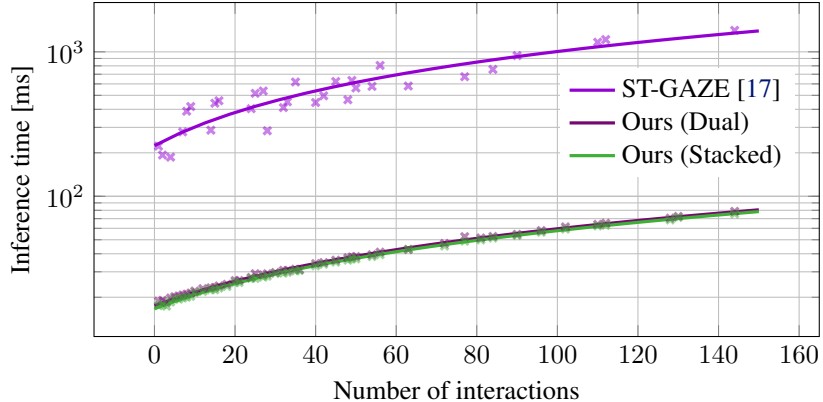

Figure 8: Model performance depends on the number of interactions for different architectures. Our variants ('Dual' and 'Stacked') have similar inference times (curves overlap) while outperforming by large margins the ST-GAZE model [17]

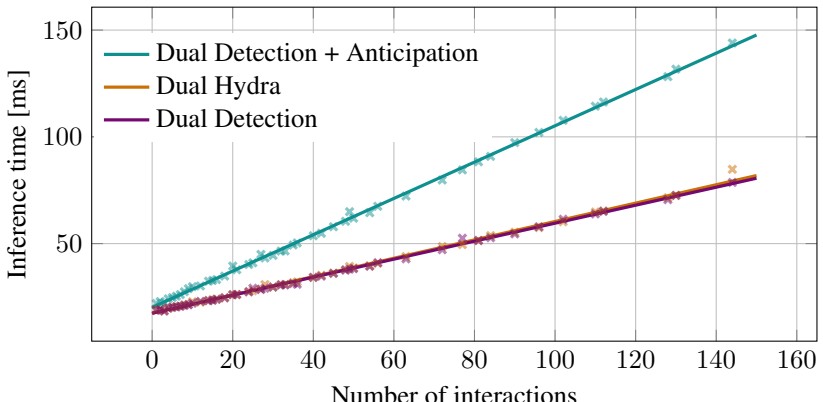

Figure 9: Model performance depends on the number of interactions for different model variants. The proposed multi-head approach allows us to detect and anticipate HOIs at multiple time horizons while maintaining a similar inference speed as the 'Dual' version (purple and dark orange curves overlap). We observe the benefit of the Hydra compared to running a specific 'Dual' transformer per detection and per anticipation.

## D    Inference time

Our model is able to run in real-time thanks to the efficient design and reduced dimensionality.

**Inference time versus the number of human-object pairs.** Due to the nature of HOIs, each interaction needs to be computed for each human-object pair existing in the scene at a given time step. Therefore, to speed up the results and parallelize the forward pass for a given video, we stack all found human-object pairs in the batch dimension. Still, we consider it necessary to observe how different models' inference speed is affected by the number of pairs in a given video. Therefore, we run 1000 executions of our model processing a given video with $I$ interactions. We implement all models reported in Fig. 8 and 9 in the same batch strategy and observe a similar tendency in the increase of the inference time for a higher number of interactions.

**Efficiency comparison with current state-of-the-art [17].** Both HOI4ABOT and [17] adopt a transformer-based architecture to comprehend the temporal relationships between the humans and objects in the scene. However, our model is designed to be efficient and to run in real-time despite having a large number of interactions, contrary to [17]. The comparison of the efficiency of both models is depicted in Fig. 8, which shows that our HOI4ABOT outperforms [17] by large margins in

Table 6: Anticipation mAP in Oracle mode.

| Method | t | mAP | Preson-wise top-5 | | | |
| --- | --- | --- | --- | --- | --- | --- |
| | | | Rec | Prec | Acc | F1 |
| STTran [15] | 1 | 29.09 | **74.76** | 41.36 | 36.61 | 50.48 |
| | 3 | 27.59 | **74.79** | 40.86 | 36.42 | 50.16 |
| | 5 | 27.32 | **75.65** | 41.18 | 36.92 | 50.66 |
| ST-Gaze [17] | 1 | 37.59 | 72.17 | 59.98 | 51.65 | 62.78 |
| | 3 | 33.14 | 71.88 | 60.44 | 52.08 | 62.87 |
| | 5 | 32.75 | 71.25 | 59.09 | 51.14 | 61.92 |
| Ours (*Dual, scratch*) | 1 | **38.46** | 73.32 | 63.78 | 55.37 | 65.59 |
| | 3 | 34.58 | 73.61 | 61.7 | 54 | 64.48 |
| | 5 | 33.79 | 72.33 | 63.96 | 55.28 | 65.21 |
| Ours (*Dual, Hydra*) | 1 | 37.77 | 74.07 | 64.9 | **56.38** | **66.53** |
| | 3 | 34.75 | 74.37 | 64.52 | 56.22 | **66.4** |
| | 5 | 34.07 | 73.67 | 65.1 | 56.31 | **66.4** |
| Ours (*Stacked, Scratch*) | 1 | 36.14 | 70.03 | 64.61 | 53.99 | 64.34 |
| | 3 | 34.65 | 73.85 | 62.13 | 54.15 | 64.77 |
| | 5 | 34.27 | 72.29 | 61.81 | 53.65 | 64.03 |
| Ours (*Stacked, Hydra*) | 1 | 37.8 | 72.05 | **65.58** | 56.23 | 66.09 |
| | 3 | **34.9** | 72.96 | **65.05** | **56.3** | 66.2 |
| | 5 | **35** | 72.86 | **65.18** | **56.36** | 66.2 |

terms of speed. Next, we list the major differences in the model design that cause our improvement. First, we do not use any additional modality to predict HOIs, compared to [17] that leverages pre-extracted gaze features to capture the human's attention. Predicting these gaze features is costly as it requires detecting and tracking each human's head in the scene, predicting the corresponding gaze per human, and matching it to the corresponding body. Thus the speed decreases considerably depending on the number of humans in the scene. Moreover, [17] also considers an initial spatial transformer that leverages all humans and objects per frame, thus [17] speed is more affected by the number of frames considered.

**Efficiency comparison of the *Hydra* HOI4ABOT.** Human intention reading requires understanding both current and future HOIs. Therefore, we develop a multi-head HOI4ABOT, called *Hydra*, that allows us to predict HOIs at different time horizons in the future through a single forward step. While Table 6 shows the benefit of our *Hydra* variant compared to training from scratch, in this subsection we focus on the benefit of efficiency. Fig. 9 shows the inference time in milliseconds depending on the number of human-object pairs across different variants. We consider the *Dual Detection* as the baseline of our HOI4ABOT model when only predicting the HOI in the present. *Dual Detection + Anticipation* is an optimized model that uses two dual transformer blocks that benefit from the same image backbone, one for HOI detection and the other for HOI anticipation in a single future $\tau = 3$. Finally, our *Dual Hydra* performs HOI detection and anticipation for $\tau = [0, 1, 3, 5]$ in a single step by using our multi-head strategy. We observe the benefit of our *Hydra* variant compared to the model ensemble, as it has a comparable speed to the single head while anticipating HOIs in three additional future horizons.

# E   Extensive comparison with variants

Our HOI4ABOT model outperforms the current state-of-the-art across all tasks and metrics in the VidHOI dataset, as shown in Tabel 6. In this section, we extend the comparison from the manuscript for the HOI anticipation for our *Dual* and *Stacked* variants, both when being trained by scratch or through the multi-head *Hydra* mode. Our results show that the *Stacked* variant obtains slightly better performance in the mAP for longer futures. We consider this marginal improvement to be motivated because of the width difference in the transformer blocks, as well as the bigger representation space from which we project when classifying the HOIs. The *Stacked* variant is based on a single self-

attention block that operates on the human windows and object windows stacked in time. Therefore, the *Stacked* transformer has double the width compared to the *Dual* variant. Given that the output of a transformer model has the same shape as its input, the obtained tokens are also wider in the *Stacked* variant. Having a bigger embedding dimension in the projected token allows the encoding of more information, which could result in better performance. However, Table 6 shows that the *Stacked* variant has a lower recall and therefore lower F1-Score. These findings might indicate that the *Stacked* variant struggles when anticipating HOIs in the videos where the interaction changes in the anticipation horizon, being more conservative in its predictions. Therefore, we consider the *Dual* variant to be optimal as it balances both precision and recall metrics across all tasks, as shown by outperforming all other models in the F1-score for the *Hydra* version.

