# OpenReview forum: "HOI4ABOT: Human-Object Interaction Anticipation for Human Intention Reading Collaborative roBOTs"
_robot-learning.org/CoRL/2023/Conference — CoRL 2023 Poster_

### Official Review · Reviewer_5Maz · 2023-07-19

**Confidence:** 4
**Originality:** Very Good
**Technical Quality:** Very Good
**Clarity Of Presentation:** Very Good
**Impact:** 3

**Recommendation:**

Weak Accept: I recommend accepting the paper, but will not argue for my recommendation if the majority of other reviewers have a different opinion.

**Review:**

Strengths:

•	The paper is well motivated in such that it proposes improving the efficiency of HOI detection and anticipation from videos for assistive robot.

•	The real-world experiment is real-time and well designed with both ‘near the cup’ and ‘away from cup’ cases considered, demonstrating the accuracy, efficiency, and flexibility of proposed method.

•	The video clearly demonstrates the real-world experiment.

Weaknesses:

•	The task has objects that are very distinct. For example, in the kitchen task, objects like laptop, cup, kitchen are different thus less ambiguity appeared. However, if some similar objects like cup and bowl(human approaches kitchen but plans to grab bowl instead of cup, and corresponding robot actions are different), kitchen and table(they are in similar direction), or even two different cups are present (and the human moves to cup1, and then moves away from cup1 to grab cup2 near the laptop), will the proposed method still be able to quickly distinguish in real time and outperform other methods (as gaze information seems to be important in these cases)?


**Quality Of The Limitations Section:**

Limitations are addressed clearly

**Questions For Rebuttal:**

1.	The Figure2 is a little bit unclear. Some labels/arrows are missing/incorrect. For example, $E^T$ is missing, there are no arrows from ViT to $E^T$ and $cls_t$, the input to box embedder seems should be $b_m^t$. And for the right part it could be better if the flow is from top to bottom instead of bottom to top.
2.	The question raised in weakness.


**Robotics Focus:**

Sufficient demonstration on hardware

**Summary Of Paper:**

This paper proposes a Human-Object Interaction (HOI) anticipation framework, HOI4ABOT, to detect and anticipate HOI from videos with less latency. HOI4ABOT, as a two-stage model, first extracts and obtains features for objects and humans per frame with Patch Merger method, and then detect and anticipate HOI using a dual transformer module with cross-attention layers. The paper also proposes a Hydra variant to predict multiple time horizon interactions. The authors conduct experiments with VidHOI dataset and show that HOI4ABOT outperforms state-of-the-art in both mAP and speed, demonstrating that HOI4ABOT is capable of accurately and efficiently detect and anticipate HOI, thus enabling robot to better assist human. The authors also conduct real-word experiments with convincing real-time performance.

**Summary Of Recommendation:**

The paper proposes to accelerate the HOI detection and anticipation from videos. The paper is well-written and overall technical sound. The literature review is thorough and experiment results are extensive and convincing. The only doubt is its performance in scenarios with more ambiguity and complexity.

---

### Official Review · Reviewer_pnXM · 2023-07-19

**Confidence:** 4
**Originality:** Good
**Technical Quality:** Fair
**Clarity Of Presentation:** Good
**Impact:** 3

**Recommendation:**

Weak Reject: I recommend rejecting the paper, but will not argue for my recommendation if the majority of other reviewers have a different opinion.

**Review:**

Strengths:
- This work tackles a difficult and important problem in robotics (anticipating human behavior)
- The work includes a fully implemented system on a robot
- The proposed methods for human activity recognition is likely to be of interest to roboticists

Weaknesses:
- The validation with the robot ("real world experiments") is not well-designed and is lacking detail (was the human in the study a member of the research team? was the study approved by a university ethics board?).  For a paper where the primary robotics innovation is incorporating the method in a system, it is important to have a strong validation to demonstrate that the method is effective and robust.
- Overall, the robotics side of the work is somewhat weak; although the authors argue the method will improve human-robot interaction, the paper does not provide sufficient evidence to support that claim, or provide experimental evidence that the method is effective across different types of interaction.


Minor points:
- the authors may want to refer to their system as a "collaborative" rather than assistive robot -- typically "assistive robotics" are robots intended to be used by disabled people.

**Quality Of The Limitations Section:**

Limitations are addressed clearly

**Questions For Rebuttal:**

The authors may wish to highlight any evidence that supports the idea that this method would improve human-robot interaction in practice.  Please also provide more detail on the experimental design.

**Robotics Focus:**

Sufficient demonstration on hardware

**Summary Of Paper:**

This paper presents a method for predicting human activity to allow a robot to anticipate and execute appropriate collaborative actions.  The system uses a vision-based machine learning model to predict the action, behavior trees to choose the robot action, and kinesthetic teaching combined with a motion planner to move the robot.  The proposed method achieves a significant speedup over prior approaches with a minor improvement in accuracy.

**Summary Of Recommendation:**

Overall, this is an interesting human activity recognition system but the robotics experiments seem incomplete.  Further validation to demonstrate that the proposed method works across tasks and users would make this a strong submission to a future venue.

---

### Official Review · Reviewer_AxBS · 2023-07-23

**Confidence:** 4
**Originality:** Good
**Technical Quality:** Very Good
**Clarity Of Presentation:** Good
**Impact:** 3

**Recommendation:**

Weak Accept: I recommend accepting the paper, but will not argue for my recommendation if the majority of other reviewers have a different opinion.

**Review:**

The paper is well-written and easy to follow. The work discusses an important problem of human-object interaction (HOI) that would be useful for robot assistance to humans and thereby improve human-robot interaction.

Strengths
- The model uses video data which is fairly ubiquitous but abstracts and combines different information that potentially improves HOI identification.
- The model uses efficient techniques to improve inference times, particularly for varying prediction horizons.
- The experiments conducted in a real-world setting with actual robots demonstrates the importance of predicting HOIs.

Weaknesses
- Definitions should be more clearly explained and more explanations could be provided to support the validity of the assumptions
- Some details on the modeling approach are unclear or missing
- Limitations of the modeling approach not discussed in detail








**Quality Of The Limitations Section:**

Additional details required

**Questions For Rebuttal:**

Responses to the following comments could help improve the paper.

- The intuition behind the model development could be explained in more detail. Why did the authors think this method/model would work?

- Similarly, the authors could also provide some intuition on why their models worked better than existing SoA methods, particularly with that of [1], which had used additional gaze data but still did not perform as well as the methods discussed in the paper. Did they use unique features that are more informative? Did their techniques better process/use the video data?

- It is difficult to understand Figure 3. What are the values, 0.3, 0.5, and 0.7 representing? The authors could also report prediction results for HOIs for this real-world experiment to understand how the model developed on offline data translates to real-world data.

- It is unclear if the context variable, $cls$ is also obtained from ViT architecture or if there was additional processing done to get $cls$. It is hard to understand the validity of the assumption in Line 145 without knowing how $cls$ was calculated.

- What do the authors consider as context? Is it the environment the interaction is happening in? Is it arrangement of objects? It would be useful to understand what is defined as context here, to understand whether assuming context to be homogeneous within a video is valid. For example, in the scenario demonstrated by the authors, a user might want to have a drink and come closer to the cup but instead change their mind and start working on their laptop. This could be considered as the context changed from "wanting a drink" to "having to work".

- The Limitations section primarily focusses on the dataset used for model development and its lack of robotics-related scenarios. While this is indeed a limitation, the authors could also expand the limitations section to discuss potential limitations of their models and methodology.


- (Line 44) The authors have stated that, “The proposed HOI4ABOT outperforms the state-of-the-art in VidHOI across all HOI tasks.” in their contributions. While this is an interesting result, I would not call this as a contribution but rather as a result/performance of the model they developed, which is the contribution.

- Figure 2 is a very detailed diagram and the authors have provided a lot of information to aid the readers understand the architecture. However, some information is unclear/could be included.
    - The arrows could be made more visible. It is hard to track the direction of data flow.
    - The text uses variables $E^t$, $B^t$ and $\hat{B}^t$ but the diagram uses only $e^t$, $b^t$ and $\hat{b}^t$. The authors can clarify what these variables are.
    - Some variables are not explicitly explained, especially on the right diagram (q, $s_m$, $S_m$, $\hat{WH}$, K, V, Q, etc.)


- Following clarifications would be useful

    - In Line 141, the authors claim their patch merger algorithm works better than [2]. Are these results included somewhere in the paper to support this claim?

    - In Line 190, the authors can provide details on the motion primitive identification. For example, how many trajectories were collected, how many motion primitives were identified in total, etc.

    - What is the train-test-val split for the dataset used in the experiments?

    - (Line 260) Is the time horizon in seconds?

    - (Line 270) What are the exact annotations in the dataset that seem appropriate for this assistive task?

    - (Line 272) Are all 20 videos of the same human?


References
1. Z. Ni, E. Valls Mascar ́ o, H. Ahn, and D. Lee. Human-object interaction prediction in 364 videos through gaze following. Computer Vision and Image Understanding, page 103741, 2023. ISSN 1077-3142. doi:https://doi.org/10.1016/j.cviu.2023.103741. URL https:  //www.sciencedirect.com/science/article/pii/S1077314223001212.
2.  J. Park, J.-W. Park, and J.-S. Lee. Viplo: Vision transformer based pose-conditioned self-loop graph for human-object interaction detection. In Proceedings of the IEEE/CVF Conference on Computer Vision and Pattern Recognition, pages 17152–17162, 2023.

**Robotics Focus:**

Sufficient demonstration on hardware

**Summary Of Paper:**

The key idea of the paper is to model human-object interaction using temporal cues from video data using transformers. A major motivation for the development of the model is real-time inference of human-object interaction detection and prediction for robotic assistance.

A major contribution of the paper is the development of a human-object interaction model that is able both detect and anticipate human object interaction better than existing models. The model is efficient and is able to provide real-time inference. Other contributions include testing on a real robot in a assistive setting by planning robot actions based on the predicted object interaction.

**Summary Of Recommendation:**

My recommendation is based on the following
- While the modeling approach uses transformers similar to some of the existing methods, the way temporal data is abstracted and how the transformer model is designed is distinct enough to improve HOI prediction.
- The model uses efficient techniques for real-time inference using temporal data suitable for real-world robotics applications.
- The demonstration of the model and corresponding robot planning on a robot assistance task motivates the necessity for HOI prediction.
- While the paper provides a lot of details on the model approach and the evaluation experiments, some important details are missing which are required to even understand the validity of some of the assumptions and for potential replication.

---

> ### Author Response · Authors · 2023-08-12
> **Rebuttal by Paper294 Authors (2/3)**
>
> _(continued)_
>
> > **Q4.** It is unclear if the context variable,  is also obtained from ViT architecture or if there was additional processing done to get CLS. It is hard to understand the validity of the assumption in Line 145 without knowing how CLS was calculated.
>
> As detailed in Line 133, our ViT backbone not only extracts patch-base features  $\mathbf{E}^t$ but also a global feature $\mathbf{cls}\_t$ that contains the overall representation of an image $\mathbf{f}\_{t}$.  This $\mathbf{cls}\_t$ token is obtained per image $\mathbf{f}\_{t}$, so we obtain $T$ global tokens $CLS = [\mathbf{cls}\_{-T},\cdots,\mathbf{cls}\_{0}]$ for a window of $T$ frames $\mathbf{V}=[\mathbf{f}\_{-T},\cdots,\mathbf{f}\_{0}]$. We are interested in the overall context of the window (for example the context might represent: a kitchen, a playground, an office, etc.). We claim that this context should be invariant in very short periods of time and be the dominant component in the $CLS$ tokens. Consequently,  we propose to use an average pooling strategy to reduce the N $\mathbf{cls}\_t$ tokens to only a single representation $\mathbf{\widehat{cls}} = AvgPool([\mathbf{cls}\_{-T},\cdots,\mathbf{cls}\_{0}])$, which is the context of the scene.
>
> > **Q5.** What do the authors consider as context? Is it the environment the interaction is happening in? Is it arrangement of objects? It would be useful to understand what is defined as context here, to understand whether assuming context to be homogeneous within a video is valid. For example, in the scenario demonstrated by the authors, a user might want to have a drink and come closer to the cup but instead change their mind and start working on their laptop. This could be considered as the context changed from "wanting a drink" to "having to work".
>
> We agree that the example given by the reviewer can be considered as the context. However, in this work, we refer to 'wanting a drink' or 'having to work' as the human intention, which is what we implicitly aim to predict through the HOIs.
> We consider the context as the overall visual setting of a scene, including elements like location, objects, and individuals present. This context, assumed consistent over a short time, aids in understanding interactions. In a 'human in a kitchen' context, the human is less likely to kick a football than to hold it. This would not be the case in a 'human on a football pitch' context. Our anticipation model leverages this context information to better predict HOIs.
>
> > **Q6.**  The Limitations section primarily focuses on the dataset used for model development and its lack of robotics-related scenarios. While this is indeed a limitation, the authors could also expand the limitations section to discuss the potential limitations of their models and methodology.
>
> Please read and consider our general response where we expand the limitations of our methodology.
>
> > **Q7.**  (Line 44) The authors have stated that, “The proposed HOI4ABOT outperforms the state-of-the-art in VidHOI across all HOI tasks.” in their contributions. While this is an interesting result, I would not call this as a contribution but rather as a result/performance of the model they developed, which is the contribution.
>
> Thank you for the comment. We agree and will include this modification in the revised version of the paper.
>
> > **Q8.**  Figure 2 is a very detailed diagram and the authors have provided a lot of information to aid the readers understand the architecture. However, some information is unclear/could be included. The arrows could be made more visible. It is hard to track the direction of data flow. The text uses variables $E^t$, $B^t$ and $\hat{B}^t$ but the diagram uses only $e^t$, $b^t$ and $\hat{b^t}$. The authors can clarify what these variables are. Some variables are not explicitly explained, especially on the right diagram (q, $s\_m$, $S\_m$, $\hat{WH}$, K, V, Q, etc.)
>
> Thank you for the constructive comment. Our revised diagram for the architecture is shown in Figure 5 of the document attached to the rebuttal comment. We highlight the data flow in the architecture and revise the variables used to better connect with the description of our framework from the manuscript. For clarification, $\mathrm{K}$, $\mathrm{Q}$, and $\mathrm{V}$ refer to the Key, Query, and Value used in the attention mechanism of transformers. $\mathrm{q}$ is a learnable spatial parameter that, quoted from our paper, 'learns the evolution of the location in time through the attention mechanism'.  $\mathrm{S\_m}$ is a spelling mistake in the figure as it should be small as $\mathrm{s\_m}$. Finally, $\widehat{\mathbf{WH}}\_n$ is the output of the human blender that represents the refined human knowledge $\mathbf{WH}\_n$ based on the output of the object blender $\widehat{\mathbf{WO}}\_m$. We will include these comments in the revised paper.
>
> _(continue)_

---

> ### Author Response · Authors · 2023-08-12
> **Rebuttal by Paper294 Authors (3/3)**
>
> _(continued)_
>
> > **Q9.**   The following clarifications would be useful:
>
> > > **Q9.1.**   In Line 141, the authors claim their patch merger algorithm works better than [2]. Are these results included somewhere in the paper to support this claim?
>
> Yes, we show these results in the ablation study Table 3: "Feature Blender: MOA".
>
> > > **Q9.2.**   In Line 190, the authors can provide details on the motion primitive identification. For example, how many trajectories were collected, how many motion primitives were identified in total, etc.
>
> Please see the answer to [_Reviewer fhCH Question 1_].
>
> > > **Q9.3.**   What is the train-test-val split for the dataset used in the experiments?
>
> We use the same splits, which are provided by the VidHOI dataset, that are used in previous works (STTRAN, VidHOI, ...). _193911_ frames are in the training set, and _22808_ in the validation set.
>
> > > **Q9.4.**   (Line 260) Is the time horizon in seconds?
>
> Implicitly yes. The time horizon is per frame, but as the dataset contains videos at one frame per second, it also refers to seconds.
>
> > > **Q9.5.**   (Line 270) What are the exact annotations in the dataset that seem appropriate for this assistive task?
>
> Some of the temporal HOI categories could be used for assistive tasks, like _push_, _pull_, _touch_, _carry_, _lift_, _hold_, _open_, _close_, or _cut_. Moreover, the spatial categories (like _next\_to_, _behind_, etc.) could be useful for tasks where the location of the human w.r.t. an object is important. A full list can be found here: https://github.com/coldmanck/VidHOI
>
> > > **Q9.6.** (Line 272) Are all 20 videos of the same human?
>
> No, 4 different humans were involved in the experiments. Each of them did 5 repetitions without any direct instruction on how to approach or grab the cup.

---

### Official Review · Reviewer_fhCH · 2023-07-27

**Confidence:** 3
**Originality:** Good
**Technical Quality:** Very Good
**Clarity Of Presentation:** Good
**Impact:** 4

**Recommendation:**

Weak Accept: I recommend accepting the paper, but will not argue for my recommendation if the majority of other reviewers have a different opinion.

**Review:**

Strengths:

-  This is a well-written paper which clearly contributes a Human-Object Interaction detection and anticipation framework.
- The novelty in the approach lies in how the approach leverages pretrained models and aligns these model outputs in a patch-merging strategy, as well as the dual transformer used for the Object Blender and Human Blender transformers.
-  Being able to anticipate HOIs is an important problem in developing assistive robots which understand human needs.
- The authors compare the method to a good number of baselines and demonstrate that their methods outperform baselines on the evaluation tasks.


Weaknesses/Areas of Improvement:

- In Section 3.2, Motion generation and task planning, details on what complex movements required decomposition into simpler movement primitives, are needed to better understand this was for the robot pouring task. The human operator guides the robot in the beverage pouring task. What was the performance and robustness of the kinesthetic teaching?
- In Table 1, in Oracle mode, the Ours (Stacked) method outperforms Our (Dual). However, in Detection mode, Ours (Dual) outperforms Ours (Stacked). Why might this be the case? Section 4.2 could provide more discussion on these differences.
- In Table 3, the results show Dual (Scratch) does not perform as well as Dual (Hydra). Section 4.2 could be expanded to elaborate more on why this is the case. Is training a Dual model specifically on a particular prediction horizon overfitting to the data?
- Figure 3 should be explained in more detail, as the finding that HOI anticipation facilitates earlier prediction of the human intention, is not immediately clear. The group within the legend should be more clearly explained.
- In Section 4.4: Real World experiments, since this method is developed towards designing assistive robots, this section would benefit from more evaluation measures of the system. How do early HOI predictions enable robots to be more effective at this task? Do the human user perceptions of this assistive robot support the need for early HOI predictions?
- The discussion section focuses on interesting limitations within the human data. It would be great for the authors to discuss limitations within the methodology, assumptions required for the nature of the videos, and the generalizability of this framework on other datasets.


**Quality Of The Limitations Section:**

Additional details required

**Questions For Rebuttal:**


- In Section 3.2, what complex movements required decomposition into simpler movement primitives? What was the performance and robustness of the kinesthetic teaching?
- In Table 1, in Oracle mode, the Ours (Stacked) method outperforms Our (Dual). However, in Detection mode, Ours (Dual) outperforms Ours (Stacked). In Table 3, the results show Dual (Scratch) does not perform as well as Dual (Hydra). Why this is the case? Is training a Dual model specifically on a particular prediction horizon overfitting to the data?
- In Section 4.4, are there other quantitative or qualitative measures to demonstrate how do early HOI predictions enable robots to be more effective at this task? Do the human user perceptions of this assistive robot support the need for early HOI predictions?


**Robotics Focus:**

Sufficient demonstration on hardware

**Summary Of Paper:**

This paper presents a transformer-based model to detect HOIs from videos. The paper contributes a Patch Merger technique for combining the information from a Visual Transformer patch encoding with an object detection bounding box to generate individual representations of all objects. Visual tokens and spatial features are stacked to form human and object temporal windows. Two cross attention transformers combine the human window information with the object window, in order to predict the human-object interaction class. To predict HOIs at multiple time horizons, the authors adopt a multihead variant.

The model was trained and evaluated on the VidHOI dataset, and performance was compared against five baselines. The results demonstrate that the approach outperforms these baselines on the detection of HOIs task. The multihead pretraining strategy outperforms training the baselines from scratch for different anticipation horizons and facilitates earlier prediction of the human intention.

**Summary Of Recommendation:**

This is a well-written paper which contributes a Human-Object Interaction detection and anticipation framework. The approach is novel and uses a transformer-based model to detect HOIs from videos, while leverages pretrained models to enhance the inputs to the transformer-based model. I would recommend acceptance with minor edits.

---

> ### Author Response · Authors · 2023-08-12
> **Rebuttal by Paper294 Authors (2/2)**
>
> _(continued)_
>
> > > **Q4.2.** Do the human user perceptions of this assistive robot support the need for early HOI predictions?
>
> In this paper, we present results that show the potential of using HOI anticipation in human assistance/collaboration, by giving the robot some insight into human intention. All results shown previously indicate the benefit of our framework, which leads to a more proactive and fluent system. That being said full-scale user experiments are further required to decisively answer this question. We leave them for future work.
>
> Other works in the literature [5, 6] have already supported the benefit of robot anticipatory behaviors for better collaborative tasks. [5] considered an experiment where a human carries car parts to a shared workspace and an anticipatory robot assembles them. Significant differences were found in the rating of the robot's contribution and commitment, as well as in the fluency metrics of C-ACT and F-DEL. [6] also showed positive benefits of perceptual anticipation in collaborative robots.
>
> Overall, we consider our work to be well-motivated in the human-robot collaborative/assistive scenario as prior evidence has similar findings.
>
> > **Q5.** Figure 3 should be explained in more detail, as the finding that HOI anticipation facilitates earlier prediction of the human intention, is not immediately clear. The group within the legend should be more clearly explained.
>
>  We acknowledge that further explanation in Figure 3 is required for deeper understanding. Figure 3 shows how much in advance (anticipation time, in seconds) each head of our _Hydra_ model (0 for detection, and 1, 3, and 5 for anticipation) is able to predict an interaction with a confidence score over a defined threshold (0.3, 0.5, or 0.7). The sooner the model is able to anticipate the interaction, the sooner the robot can start executing some of the movements, thus reducing the waiting time of the human. To better illustrate the advantages of using our model, we propose to substitute Figure 3 in the revised manuscript with the new Figure 3 in the attached document, which explicitly shows the time a human waits until the robot starts to serve the drink (the pouring movement). Effectively, by adding the duration of the robot pouring motion, we can obtain the overall waiting time for the drink to be poured.
>     We can observe in both figures that when the model is only detecting the HOI ($\tau_a=0$), the robot starts reacting later. In the anticipation ($\tau_a=\{1,3,5\}$), our robot becomes proactive and starts assisting earlier (up to 3 seconds). The figure shows a slight reduction in the waiting time when reducing the confidence threshold in the prediction: to be more confident in the human intention the robot waits more. In addition, we observe only a slight decrease in the waiting time for different anticipation horizons ($\tau_a=\{1,3,5\}$). This subtle variation might be caused because of the dataset limitation pointed out in Section 3.
>
> > **Q6.** The discussion section focuses on interesting limitations within the human data. It would be great for the authors to discuss limitations within the methodology, assumptions required for the nature of the videos, and the generalizability of this framework on other datasets.
>
> Please refer to the general comment for the response.

---

### Author Response · Authors · 2023-08-12
**General Response (1/3)**

We appreciate all the valuable suggestions from the reviewers. They considered our work to tackle a significant problem in robotics, be well-supported, and clearly presented. Moreover, all reviewers positively remarked on the performance and efficiency of the proposed solution and real-world robotic experiments. We provide a general response next to showcase the evidence that detecting and anticipating HOIs improves collaboration, as well as we extend the limitation discussion of our framework. Then we respond to each reviewer's detailed questions separately.

_As it is not possible to attach files to the general comment, the files (including the videos and figures for the rebuttal) can be found in the rebuttal to each reviewer._

**Regarding the evidence that detecting and anticipating HOIs improve collaboration.**

We extend our evaluation of the framework in real-world experiments with additional quantitative and qualitative experiments and metrics.

First, we assess the human waiting time until the robot proceeds to serve. In Figure 3 of the attached document, we showcase the quantitative benefit of our approach by considering the absolute time a human waits to be served (serving is considered until the robot starts pouring). The results indicate that our robot behaves proactively when anticipating HOIs and therefore reduces the time to wait until a drink is poured, compared to the reactive behavior observed if the robot is only detecting HOIs. This is further elaborated in _(Reviewer fhCH, Question 5)_.

Secondly, we measure the effectiveness of our robot pouring a drink in our real-world trials by considering the success rate of the pouring task in _20_ new real-world experiments. Similarly, like in previous experiments, four lab members were instructed to approach the robot and grab the cup. Each person did 5 repetitions. Our framework correctly executes the pouring task in _17_ out of _20_ executions, resulting in a success rate of _85\%_.

Thirdly, we extend our evaluation by considering well-established metrics in human-robot collaboration [4]. [4] proposes to evaluate human-robot fluency in the joint task by considering four objective metrics. _Human Idle Time_ (H-IDLE) and _Robot Idle Time_ (R-IDLE) are proposed to evaluate the percentage of the total task time that the respective agent is not active, which reflects the team coordination and the inefficiency of the agent in the task. _Concurrent Activity_ (C-ACT) measures the percentage of total task time in which both agents are active concurrently (the action overlap between different members). A higher  C-ACT indicates a team that is well synchronized. _Functional Delay_ (F-DEL) measures the delay experienced by the agents immediately after completing an activity: the percentage of total task time between the completion of one agent's action and the beginning of the other agent's action. A negative F-DEL indicates that actions are overlapping and implies an efficient use of team members' time. Figure 4 in our attached document summarizes the average Objective fluency metrics across our pouring experiments. The results indicate that HOI anticipation allows for substantially better human-robot coordination and efficiency of each other's time, thus making the task more fluent.

Finally, we provide additional demonstrations of the system's performance in more ambiguous scenes as well as a different assistive scenario. Please see the attached videos, which encompass:
- **Video 'Ambiguous_Pour'.** The robot pours a drink into the cup held by the human. We consider this as an extension of our initial video, as the robot generates a movement conditioned by the 'cup' instance the human is holding among different options. In the video other interactions are also shown, which indicate the generability of our approach. The robot then performs a second pouring.
-  **Video 'Distingusihing_Objects_and_People'.** This video shows three different scenarios which depict the ability of the model to distinguish between objects and people instances in a living room and kitchen scenario. For visual simplification, we only show the detected HOIs.
- **Video 'Object_Handover'.** We define a new scenario where the robot passes over a bottle or a book depending on the predicted HOIs. If the robot observes that the human approaches the kitchen island with the cup, it passes over the bottle so the human can pour water. On the contrary, if the human is carrying a backpack while approaching, the robot passes over the book. The attached video showcases this scenario and the ability of our model to execute different movements based on the interaction. Our motion generation is not limited by the framework, and more complex movements can easily be taught to the robot.

---

> ### Author Response · Authors · 2023-08-12
> **General Comment (2/3)**
>
> **Regarding the limitation section.**
>
> **Limitations in methodology.**
> Some possible rooms for improvement are as follows. Firstly, our model is limited to _557_ HOIs categories from the VidHOI dataset. An open vocabulary model generalizes better to new interactions, but this requires much more diverse training data, which currently is not available.
> Secondly, our method predicts potential HOIs for every human-object pair in a scene. Although we use a batch strategy, our approach's efficiency can be influenced by the number of interaction pairs (see Suppl Material Figure 2). An early pair rejection system might improve this.
>
> **Assumptions required for the nature of the videos.**
> We assume in our real experiments that the objects present in the scene are sufficiently visible, so the object detection algorithm can recognize them. We also assume standard light conditions and camera setting such that the images are not under/overexposed.
>
> **Limitations in  the generalizability of this framework on other datasets.**
> Well-known available datasets for HOI detection in images are HICO-DET and V-COCO. Despite our model can work in image-based datasets, we would not exploit its full benefits as the transformer is a sequence-based model. As we are interested in a robot that can anticipate the human intention to proactively assist a human, image-based datasets are not suitable as there is no information about the human in the next frames.
> Therefore, we are interested in HOI datasets in videos. The most used are Action Genome (AG) and VidHOI. AG has been used for HOI detection but does not track the humans and objects in the videos. Thus, the dataset is unsuitable for anticipation: one can not associate future interactions with the previous human. Additionally, this tracking information is necessary for our model to construct the windows. In conclusion, VidHOI is the only dataset available that satisfies the requirements for the task.

---

> ### Author Response · Authors · 2023-08-12
> **General Comment (3/3)**
>
> **References used during the Rebuttal.**
>
> _[1] Esteve Valls Mascaró, Hyemin Ahn, Dongheui Lee; Proceedings of the IEEE/CVF Winter Conference on Applications of Computer Vision (WACV), 2023, pp. 6048-6057_
>
> _[2] Kumar Ashutosh, Rohit Girdhar, Lorenzo Torresani, Kristen Grauman; Proceedings of the IEEE/CVF Conference on Computer Vision and Pattern Recognition (CVPR), 2023, pp. 23066-23078_
>
> _[3] D. Roy and B. Fernando, "Action anticipation using latent goal learning," 2022 IEEE/CVF Winter Conference on Applications of Computer Vision (WACV), Waikoloa, HI, USA, 2022, pp. 808-816, doi: 10.1109/WACV51458.2022.00088._
>
> _[4] G. Hoffman, "Evaluating Fluency in Human–Robot Collaboration," in IEEE Transactions on Human-Machine Systems, vol. 49, no. 3, pp. 209-218, June 2019, doi: 10.1109/THMS.2019.2904558._
>
> _[5] G. Hoffman and C. Breazeal, “Cost-based anticipatory action-selection for human-robot fluency,” IEEE Trans. Robot., vol. 23, no. 5, pp. 952–961, Oct. 2007_
>
> _[6] G. Hoffman and C. Breazeal, “Effects of anticipatory perceptual simulation on practiced human-robot tasks,” Auton. Robots, vol. 28, no. 4, pp. 403–423, May 2010_
>
>
> _[7] Z. Ni, E. Valls Mascaro, H. Ahn, and D. Lee. Human-object interaction prediction in videos through gaze following. Computer Vision and Image Understanding, page 103741, 2023. ISSN 1077-3142. doi:https://doi.org/10.1016/j.cviu.2023.103741._
>
> _[8] Y. Cong, W. Liao, H. Ackermann, B. Rosenhahn, and M. Yang. Spatial-temporal transformer for dynamic scene graph generation. In 2021 IEEE/CVF International Confer357 ence on Computer Vision (ICCV), pages 16352–16362, Los Alamitos, CA, USA, oct 2021. IEEE Computer Society. doi:10.1109/ICCV48922.2021.01606._

---

### Decision · Program_Chairs · 2023-08-30

**Decision:**

Accept (Poster)

**Comment:**

The reviewers commend the paper for being well-written and organized. The proposed method to detect human-object interaction using transformers with videos was generally well received by reviewers. Reviewer pnXM brings up valid points which could benefit being addressed in the final paper, including the validation study only being conducted with a small number of lab members. I encourage the authors to further address any remaining points raised in the reviews for the final version of the paper.